# Medicinal Plants and Mushrooms with Immunomodulatory and Anticancer Properties—A Review on Hong Kong’s Experience

**DOI:** 10.3390/molecules26082173

**Published:** 2021-04-09

**Authors:** Grace Gar-Lee Yue, Clara Bik-San Lau, Ping-Chung Leung

**Affiliations:** 1Institute of Chinese Medicine, The Chinese University of Hong Kong, Shatin, New Territories, Hong Kong, China; graceyue@cuhk.edu.hk (G.G.-L.Y.); claralau@cuhk.edu.hk (C.B.-S.L.); 2State Key Laboratory of Research on Bioactivities and Clinical Applications of Medicinal Plants, The Chinese University of Hong Kong, Shatin, New Territories, Hong Kong, China

**Keywords:** medicinal herbs, medicinal mushrooms, *Coriolus versicolor*, immunomodulatory effects, cancer

## Abstract

The immune modulating effects of selected herbs deserve careful studies to gain evidence-based support for their further development. We have been working hard on many items of medicinal herbs to gain insight into their immunomodulatory effects relevant to cancer treatment in particular, while infection control is not excluded. Nine of them have been selected to give the results of our exploration on their biological, particularly immunomodulatory activities. Since Hong Kong people especially favor one medicinal mushroom, viz. *Coriolus versicolor*, a number of clinical trials using *Coriolus* for cancer-related studies are included in this review. While immune modulation platforms are being built for relevant studies, a brief account on the research targets and related procedures are given.

## 1. Introduction

Special plants have enjoyed the genuine trust of people around the world as being good for health, sometimes with special emphases on longevity. Ancient China, in particular, has lists of choice plants to maintain health.

Since the Han Dynasty (206 B.C.–220 A.D.) remarkable documentations are available. One special area of intense interest and wisdom is related to infections and epidemics: from clinical presentations and treatment choices to public health measures. When bodily defense was mentioned in those days, although no assumption needs to be raised about immunological activities, it is also clear that advocates’ and narrators’ line of thought could be very near the modern concept of immunology at its most primitive level [1,2].

Herbal formulations have been listed for the treatment of various infections in epidemics, their indications of use, and ways of preparation, which are handed down to today to facilitate therapeutic and preventive measures. The rich collection of plant items used for infection control have been studied in the laboratory and clinically along the modern directions of immunological activities—first with reference to infection as described in the old days and later diversifying into other immunological areas, notably cancer.

This review will first briefly explore medicinal herbs in traditional Chinese medicine that have been studied in modern research platforms to identify their unique influences in some specific cellular and serological tests related to immunological activities. Some of these herbs have been selected in our home-ground to serve our aspirations on cancer treatment and epidemics.

## 2. The Hong Kong Experience: Immunological Properties of Medicinal Herbs Used Against Cancer

Numerous preclinical studies have demonstrated the antitumor activities of herbal medicines and their mechanisms of action. They may either directly inhibit the growth of tumors or indirectly exert an antitumor effect by enhancing the body’s immune function [1]. Over the past 15 years or more, our institute has carried out many preclinical investigations on the antitumor and/or immunomodulatory activities of various Chinese medicinal herbs and mushrooms (Table 1). For instance, in collaboration with the Memorial Sloan Kettering Cancer Center at New York, USA, the immunomodulatory activities of five medicinal herbs/mushrooms, namely *Astragalus membranaceus*, *Coriolus versicolor*, *Curcuma longa*, *Echinacea purpurea*, and *Grifola frondosa*, have been investigated using various types of preclinical models [3]. Promising results on *Astragalus* [4,5] and *Curcuma longa* [6,7] led to further investigations on these herbs in our institute. Furthermore, a panel of herbs, which are commonly prescribed to cancer patients according to traditional Chinese medicine (TCM) theory, have also been examined for their in vitro effects on cancer cells and/or on lymphocytes in our early screening program. Based on the results, seven medicinal herbs or mushrooms, namely *Acanthopanax senticosus*, *Agaricus blazei*, *Curcuma longa*, *Ganoderma lucidum*, *Ganoderma sinense*, *Hedyotis diffusa*, and *Scutellaria baicalensis*, with particular reference to their antiproliferative effects in cancer cells or immunomodulatory properties, have been further investigated for their mechanisms of action. In addition, since the establishment of our Partner State Key Laboratory in 2010, research collaboration with State Key Laboratory of Phytochemistry and Plant Resources in West China, Kunming Institute of Botany, Chinese Academy of Sciences, has been focusing on several natural products, orchids as well as Yunnan wildly grown mushrooms. One of the edible mushrooms, *Rubinoboletus ballouii,* was shown for its immunomodulatory activities for the first time in our studies [8,9].

Hence, in this part of the review, the preclinical findings of the above-mentioned nine medicinal botanicals from our previous studies have been summarized. The clinical prospects of these individual botanicals, especially their potentials in cancer management with special focus on immunomodulation, have also been discussed.

### 2.1. Agaricus blazei

The Basidiomycete fungus *Agaricus blazei* (Agaricaceae family) is an edible mushroom, native to southern Brazil and was introduced to East Asia in the 1950s. It is commonly cultivated in Japan, China, and Brazil [10,11]. Our study firstly demonstrated the potent tumor-selective growth inhibitory activity of an ethanol–water extract of *A. blazei* against human leukemia NB-4 cells with the concentration that was required for 50 % inhibition (IC_50_ value) of 82.2 µg/mL. This extract at 500 mg/kg/day also inhibited the progression of NB-4 tumors in athymic nude mice by inducing apoptosis, which was indicated by DNA fragmentation in the tumors [12]. In view of the clinical studies on *A. blazei*, there were several other studies testing the efficacy of the water extract in patients with acute non-lymphoblastic leukemia [13], gynecological cancer [14], and prostate cancer [15]. Other clinical studies on the safety of *A. blazei* have also been conducted in cancer patients [16]; however, the dosages of the *A. blazei* extracts in these studies have not clearly reported. On the other hand, health supplements containing *A. blazei* were also shown to be effective in improving symptoms in inflammatory or allergic conditions [11].

### 2.2. Ganoderma lucidum and Ganoderma sinense

We have made attempts to study the antitumor and immunomodulatory effects of *Ganoderma* species, including different parts of the fruiting bodies, as demonstrated in human breast and liver cancer cells at effective concentrations (100–400 µg/mL) [17,18] and in S-180 sarcoma-bearing mice (at dosage 200 mg/kg) [19]. The highlights of these studies were the superior efficacies of the *G. lucidum* stipes, whose extract could exert higher cytotoxicity in cancer cells [18] and stronger immunomodulatory activities in terms of enhancing the proliferative responses and the cytokines interferon-γ (IFN-γ), interleukin-4 (IL-4) and interleukin-6 (IL-6) productions of the spleen lymphocytes of tumor-bearing mice [19], when compared with the efficacies of the *G. lucidum* pileus. On the other hand, the chemical and immunomodulatory activities of another *Ganoderma* species listed in Chinese Pharmacopoeia, *G. sinense*, were also investigated in our laboratory. The polysaccharide-enriched fraction of *G. sinense* hot water extract (50–400 µg/mL) was shown to have potent immunostimulating activities in human peripheral blood mononuclear cells (PBMCs) and monocyte-derived dendritic cells in terms of proliferative responses and cytokine productions [20]. Moreover, a series of polysaccharides were isolated from *G. sinense* and were characterized in detail [21,22,23]. Polysaccharide GSP-6B (molecular mass of 1.86 × 10^6^ Da), with a backbone of (1→6)-linked-β-D-glucopyranosyl residues and branches at the O-3 position of every two sugar residues along the backbone, was shown to induce the release of interleukin-1β (IL-1β) and tumor necrosis factor-α (TNF-α) in PBMCs, with an effective range of 0.003 to 100 µg/mL [21]. While a protein-bound polysaccharide (GSP-4, at 200–800 µg/mL), with a molecular weight of 8.3 × 10^5^ Da and the sugar residues t-; 1,3-; 1,4-; 1,6-; 1,3,4-; and 1,3,6-linked Glcp, t-linked Galp, and 1,6-linked Manp, was shown to modulate nitric oxide (NO), TNF-α, and IL-6 production in macrophage RAW264.7 cells. GSP-4 also enhanced the expression of inducible NO synthase mRNA in a dose-dependent manner [22]. Additionally, polysaccharide GSP-2 (at 30 µg/mL), with a molecular size of 32 kDa, a backbone of (1R4)– and (1R6)–Glcp, bearing terminal- and (1R3)–Glcp side-chains at the O-3 position of (1R6)–Glcp, was also isolated from *G. sinense* and was shown to induce the proliferation of mouse splenocytes targeting B cells only [23]. In addition, the potential antitumor small molecules in the stipe of *G. sinense* were identified using LC-QTOF-MS with a multivariate statistical tool [24].

On the other hand, five randomized controlled trials comparing the efficacy of *G. lucidum* medications to active or placebo control in cancer patients have been included in a systematic review and meta-analysis, in which the authors concluded that *G. lucidum* could be administered as an alternative adjuvant to conventional treatment in consideration of its potential of enhancing tumor response and stimulating host immunity [25]. In particular, in the clinical trial of breast cancer patients, *G. lucidum* spore powder treatment (1000 mg three times a day for 4 weeks) was shown to have significant improvements on physical well-being and fatigue. These patients also reported less anxiety and depression and better quality of life [26]. Nevertheless, there has been no clinical trial on *G. sinense* for cancer patients up till now.

### 2.3. Rubinoboletus ballouii

*Rubinoboletus ballouii* (RB) of the family Boletaceae, is an edible mushroom commonly found in the forest of Yunnan Province, China. Our study showed that the ethanol extract exerted anti-inflammatory effects using PBMCs. With bioassay-guided fractionation using high-speed counter current chromatography, two immunosuppressive compounds, 1-ribofuranosyl-s-triazin-2(1H)-one and pistillarin were isolated and characterized [8]. These compounds (at 20 and 100 µM) were shown to inhibit phytohemagglutinin-stimulated PBMC proliferation and inflammatory cytokine (TNF-α, IL-1β, IL-10, and IFN-γ) productions in a concentration-dependent manner. On the other hand, bioactive polysaccharides, which contain 80.6% (*w/w*) of neutral sugars including D-fucose, D-mannose, D-glucose, and D-galactose (1.7:1.4:1.0:1.8), and 12.5% (*w/w*) of proteins, were isolated from the water extract of RB and were evaluated for their immunomodulatory activities. Our results showed that the activation of monocyte-derived dendritic cells by RB polysaccharides (50 µg/mL) was mediated through the nuclear factor-κB (NF-κB) pathway via toll-like receptor-4 (TLR-4). Furthermore, RB polysaccharides at dosages of 40 and 200 mg/kg/day could restore the suppressed hematopoietic stem cell activity and upregulate the function of T helper cells in mice treated with cyclophosphamide [9]. These findings suggested that polysaccharides from *Rubinoboletus ballouii* exhibited immunostimulatory effects and have the potential to be developed as a health supplement for combined use with chemotherapeutic agents.

To date, there were only a few clinical reports on the poisoning activities of *Boletus* mushrooms [27,28]. While many natural compounds isolated from *Boletus* mushrooms were reported, such as those with apoptotic effect in cancer cells [29], antioxidant effect [30], and a liver-protective effect in mice [31]. Nevertheless, the immunomodulatory effect of RB is worth further investigating in the future.

### 2.4. Acanthopanax senticosus

The dried root and rhizome or stem of *Acanthopanax senticosus* (AS, also known as Siberian ginseng or Ciwujia in Chinese) is suggested to have immune-regulating function [46], such as the amelioration of inflammation and fatigue, as well as inhibition of tumor progression. In our laboratory, the potencies of AS aqueous extract (25–400 µg/mL) and ethanol extract (25–100 µg/mL) in regulating cytokine productions (IL-2, IFN-γ, and TNF-α) from PBMCs were determined. Our results suggested that AS aqueous extract, rather than ethanol extract or individual components (isofraxidin, syringin, or eleutheroside E), should be used in order to achieve its immunomodulatory properties [32]. In addition, the clinical efficacy of an AS-containing injection (Aidi injection) in advanced non-small-cell lung cancer patients was evaluated [47,48]. A clinical trial using AS alone has not yet been reported.

### 2.5. Astragalus membranaceus

Radix Astragali (Huangqi), the dried root of *Astragalus membranaceus* and *Astragalus mongholicus*, is a common Chinese herbal medicine as well as an ingredient of soup in Chinese cuisine. Being one of the immunomodulators studied in an National Institutes of Health (NIH)-supported research project, our previous study demonstrated that, when mice were immunized with immunogens (globo H-keyhole limpet hemocyanin (KLH) and GD3-KLH) present on the cell surface of cancer mixed with 95% ethanol extract of *Astragalus* (0.5–2 mg), significant immunological adjuvant activity could be observed [3]. Subjected to further fractionation, several flavonoids isolated from Radix Astragali, including astragalosides II (20–100 µg) and IV (20–100 µg), have been shown to have significant adjuvant activity in terms of antibody responses against cancer antigens and KLH [4]. On the other hand, the quality, purity, and uniformity of commercial Radix Astragali were assessed using chemical and genetic analyses, namely ion trap LC-MS and nuclear ribosomal DNA barcoding sequence analyses, in our collaborator laboratory in New York, USA [5]. In regard to the clinical trials on Radix Astragali alone, the majority of them focused on cardiovascular or kidney diseases [49,50]. In addition, the efficacies of *Astragalus*-containing Chinese herbal combinations have been evaluated in many clinical trials of advanced non-small-cell lung cancer patients [51]. Furthermore, the effects of *Astragalus* polysaccharides in combination with chemotherapeutics have also been examined in advanced non-small-cell lung cancer patients (250 mg/day, intravenous infusion) [52], and advanced pharyngeal or laryngeal squamous cell carcinoma patients (500 mg/day, intravenous infusion) [53].

### 2.6. Curcuma longa

Turmeric, the dried rhizome of the plant *Curcuma longa* (Zingiberaceae family), is a commonly used medicinal herb as well as a cooking herb in South or Southeast Asia and China for centuries. Thousands of preclinical studies reported the multifunctional bioactivities of curcumin [54], one of the major constituents of turmeric, including anticancer and anti-inflammatory effects in various cancer cells and animal models [55,56]. Apart from the collaborative study with Memorial Sloan Kettering Cancer Center in the U.S. on the immunological effects of turmeric, our institute has conducted a series of studies on the antitumor and anti-angiogenic activities of turmeric for a decade. We firstly confirmed the more potent antiproliferative activities of curcumin in turmeric ethanol extract than that of curcumin alone in colon cancer cells and endothelial cells [7]. We also reported the immunomodulatory activities (proliferation and TNF-α, IFN-γ productions) of turmeric polysaccharides (at 200 µg/mL) in PBMCs [6], and the anti-angiogenic activities of aromatic-turmerone in human endothelial cells, HMEC-1, (effective concentrations at 4.6–9.2 µM) and in vivo angiogenesis models (zebrafish and Matrigel plug in mice, effective concentration at 25 µg/mL) [33]. Meanwhile, we tested our hypothesis that the poor bioavailability of curcumin could be improved when it is used as a whole in turmeric extract. Results from both an in vitro Caco-2 cell monolayer study [34] and an in vivo pharmacokinetic study in mice [35] provided evidence that supported our hypothesis. Furthermore, the superior antitumor effects of turmeric extract (6.95 µg/mL, which contain 1.3 µg/mL curcumin) in colon-tumor-bearing mice than curcumin (1.3 µg/mL) alone was also demonstrated in our subsequent study [35]. On the other hand, we revealed that the antitumor activities of turmeric extract (400 mg/kg) plus bevacizumab (0.4 mg/kg) were comparable to those of the first-line chemotherapeutic, 5-fluorouracil, leucovorin, and oxaliplatin (FOLFOX), and without the hematological side effects caused by FOLFOX [36]. With higher levels of suppressed anti-angiogenesis and increased apoptosis found in tumors of turmeric-extract-treated mice than those of curcumin-treated mice, our findings from these studies provide strong evidence for the importance of turmeric ethanol extract (containing absorbable curcumin and other active components) in treating colon cancer. Last, but not least, the antimetastatic efficacies of turmeric extract were elucidated in our recent studies. Turmeric extract (200–400 mg/kg) was shown to decrease colon cancer cell migration and epithelial–mesenchymal transitions through multiple pathways, such as cofilin, focal adhesion kinase/phospho-Src (FAK/p-Src), protein kinase B (AKT), extracellular-signal-regulated kinase (Erk), and signal transducer and activator of transcription 3 (STAT3) pathways. It also decreased colon tumor burden and metastasis and enhanced immunity through T cell stimulation in a syngeneic orthotopic colon tumor mouse model [37]. In addition, turmeric extract (200 mg/kg) treatment in colon-patient-derived xenograft mice could reduce tumor progression, inhibit metastasis via modulating molecules involved in the Wnt and Src pathways, epithelial–mesenchymal transition (EMT)- and epidermal growth factor receptor (EGFR)-related pathways [38]. From these recent studies, the role of turmeric extract in regulating the colon tumor microenvironment has been revealed for the first time.

In the last two decades, over 75 clinical trials of curcumin/turmeric on chronic inflammatory diseases or cancers have been conducted [57,58,59,60]. The majority of the trials used curcumin alone as intervention, while some of them used mixtures of curcumin plus curcuminoids and/or turmeric oil. In general, these clinical trials have confirmed that curcumin/turmeric supplements can provide symptomatic relief, as well as improve tumor markers and other parameters of various cancer conditions [59].

### 2.7. Hedyotis Species

*Hedyotis (Oldenlandia) diffusa* is one of the herbs most commonly used in Chinese herbal medicines for treating cancer. Various studies demonstrated its anticancer effects in in vitro and in vivo models [61,62]. Our previous study developed a simple thin-layer chromatographic method to distinguish between *H. diffusa* and its allied species *H. corymbosa* [39]. In the subsequent studies, hedyotiscone A (25–50 µg/mL) isolated from *H. corymbosa* was found to reverse multidrug resistance in hepatoma cells by downregulating P-glycoprotein expression and *MDR1* gene expression [40]. Additionally, 4-vinylphenol isolated from *H. diffusa* water extract was firstly proven for its anti-angiogenic activities in Matrigel and tumor-bearing mouse models (at 0.2–2 mg/kg) as well as in human endothelial cells (at 20–40 µg/mL) via the phosphoinositide 3-kinase/protein kinase B (PI3K/AKT) pathway [41]. Despite many reports on the preclinical pharmacological activities of *H. diffusa* and the high usage of this herb in cancer patients clinically [63,64], proper clinical trials on this herb alone have not been reported. In contrast, the *H. diffusa*–containing formula was tested in chronic endometritis patients [65]. Further prospective clinical trials in cancer patients may be warranted in order to confirm the efficacies of this well known and commonly used “folk” anticancer herb.

### 2.8. Scutellaria barbata

Herba Scutellaria barbatae (SB), the whole plant of *Scutellaria barbata*, is commonly found as a functional component in numerous TCM prescriptions. SB is a popular herbal medicine, which has been demonstrated to have valuable therapeutic outcomes on anti-inflammation, detoxification, elimination of blood stasis, improvement of blood circulation, and anti-swelling, etc., under traditional Chinese medicine theory [66]. An active compound, designated pheophorbide a, has been isolated from SB in our earlier studies. Pheophorbide a was shown to possess photodynamic activity with apoptosis induction in human hepatocellular carcinoma (at 0.3 mg/kg) [42,43] and breast tumor (at 2.5 mg/kg) [44] mouse models. In addition, our recent study revealed the antitumor and antimetastatic effects of SB (615 mg/kg) and its chemical marker scutellarin (7 mg/kg) in a colon tumor mouse model [45]. In the same study, we also showed the regulation of cancer-metastasis-related proteins, such as e-cadherin, tetraspanin 8 (Tspan 8), and C-X-C motif chemokine receptor 4 (CXCR4), exerted by SB water extract.

In view of the clinical trials on SB, an SB aqueous extract (BZL101) has been tested in two phase I clinical trials of metastatic breast cancer [67,68]. Results showed promising clinical evidence of anticancer activity in women with metastatic breast cancer, with good safety profile, and it is well tolerated [68]. Nonetheless, similar to *Hedyotis diffusa*, *Scutellaria barbata* is always used in combination with other herbs; thus, no further clinical trials on this herb have been reported recently.

## 3. The Hong Kong Experience on Clinical Applications

Research related to the immunological effects of medicinal herbs in Hong Kong has been influenced by the behavior of the general public in pursuit of better health. Inhabitants of the island city are fond of taking health supplements in the form of proprietary medicine originated from China and Hong Kong, which contains a rather limited number of herbs exemplified by those discussed in the last section. They give special favor to medicinal mushroom such as *Ganoderma*. Since the 1970s, a new fungal fruit body called *Coriolus versicolor* (Yun Zhi) has attracted extensive attention, both in the market and among health professionals. A strong belief that *Coriolus* and its extract polysaccharopeptide (PSP), could help cancer patients with regard to survival and alleviation of adverse effects during chemotherapy has developed.

The immunomodulating effects of PSP have been studied intensively from animal experiments to clinical trials, from in vivo studies to in vitro analysis, and from cell cultures to molecular and genomic explorations. The laboratory results greatly facilitated marketing, gaining substantial social and economic profits [69].

Apart from anticancer effects, PSP was also found to be antiviral (human immunodeficiency virus, HIV) and hepatoprotective. Other studies showed significant analgesic effects of PSP on acute and chronic inflammatory pain induced in rats. Since this analgesic effect could be antagonized by the cerebral intraventricular injection of anti-IL-2 serum, the central analgesic activity of PSP may be mediated by the IL-2 receptor. Since the immune system and nervous system provide two major forces in the maintenance of the internal equilibrium of the whole body, through cellular and molecular networks, PSP is assumed to be influential in this process [69] (Figure 1).

*Salvia miltiorrhiza* (Danshen) has been one of the most commonly used medicinal herbs in traditional Chinese medicine, very much due to its vascular promotion activities, it has been used coupled with *Coriolus* in experimental studies and clinical trials. A number of studies have been completed in early 2000, and three examples are presented as follows.

### 3.1. Testing the Immunomodulatory Effects of Coriolus versicolor and Salvia miltiorrhiza Capsules in Healthy Subjects—A Randomized, Double-Blind, Placebo-Controlled, Crossover Study

One hundred healthy subjects were recruited to take Yun Zhi (50 mg/kg body weight) plus Danshen (20 mg/kg body weight) or placebo capsules daily for four successive months and, after a 2 month wash-out period, crossover to take placebo or Yun Zhi plus Danshen capsules for another four successive months. Flow cytometry was used to assess the lymphocyte subtypes and concentration of T helper (Th) cell cytokines in culture supernatant. Gene expression of cytokines and cytokine receptors of peripheral blood mononuclear cells (PBMCs) was analyzed by a cDNA expression array. Results showed that regular oral consumption of Yun Zhi–Danshen capsules could significantly elevate PBMC gene expression of interleukin (IL)-2 receptor, increase the percentage and absolute counts of T helper cell and ratio of CD4+ (T helper)/CD8+ (T suppressor and cytotoxic T) cell, and significantly enhance the ex vivo production of typical Th1 cytokine interferon-gamma from PBMC activated by phytohemagglutinin and lipopolysaccharide (all *p* < 0.005). Such consumption had no adverse effects on the liver and renal functions and the biochemical profile [70].

### 3.2. Coriolus and Salvia for Patients Suffering from Nasopharyngeal Carcinoma

Twenty-seven patients with histologically proven nasopharyngeal carcinoma were recruited to take Yun Zhi (3.6 g daily) and Danshen (1.4 g daily) in the form of 12 combination capsules (TCM group) or placebo (12 capsules) daily for 16 weeks, respectively. Flow cytometry was used to assess the percentages and absolute counts of human lymphocyte subsets in whole blood. Plasma concentration of soluble interleukin-2 receptor and soluble tumor necrosis factor receptor 2 were measured by enzyme-linked immunosorbent assay (ELISA). Ex vivo production of tumor necrosis factor-alpha, interleukin (IL)-6, and IL-10 in the whole blood assay culture supernatant was measured by ELISA.

Results showed that the decreases in percentage and absolute count of T lymphocytes in the treatment group were less than those in the placebo group after they took the capsules for 16 weeks (both *p* < 0.05). Furthermore, the decreases in the absolute count of T suppressor cells, cytotoxic T lymphocytes, and T helper cells in the treatment group were significantly lower than those in the placebo group after they took the capsules for 16 weeks (both *p* < 0.05) [71].

### 3.3. Coriolus and Salvia for Breast Cancer Patients after Completion of Treatment

Bodily discomfort and fatigue are experienced by cancer patients undergoing treatment: from surgery and radiotherapy to chemotherapy, because of suppressed immunological functions. Eighty-two patients who completed treatment for breast cancer were recruited to take Yun Zhi (50 mg/kg body weight, 100% polysaccharopeptide (PSP)) and Danshen (20 mg/kg body weight) capsules every day for a total of 6 months. Ethylenediaminetetraacetic acid (EDTA) blood samples were collected every 2 months for the investigation of immunological functions. Flow cytometry was used to assess the percentages and absolute counts of human lymphocyte subsets in whole blood. Plasma level of soluble interleukin-2 receptor (sIL-2R) was measured by enzyme-linked immunosorbent assay (ELISA). Results showed that the absolute counts of T-helper lymphocytes (CD4+), the ratio of T-helper (CD4+)/T suppressor and cytotoxic lymphocytes (CD8+), and the percentage and the absolute counts of B-lymphocytes were significantly elevated in the treatment group. The plasma slL-2R concentration was also significantly decreased (all *p* < 0.05) among these patients [72].

Cancer patients were given the *Coriolus* and *Salvia* capsules after they completed standard treatment. They were expected to enjoy better quality of life, and more specifically, we critically looked at the immunological effects, which would offer quantitative data on clinical benefits. The follow-up periods were short, which would not allow longer-term studies on clinical benefits.

## 4. Common Target Areas That Investigators Used to Study the Bioactivities Related to Immunological Properties of Medicinal Plants:

### 4.1. Organs Related to the Provision of Relevant Cells and Cellular Activities: [73]

bone marrow,thymus,lymph nodes and lymphatic system, andspleen.

Hypertrophy of these organs with increased cell production must be the most accepted indications of a medicinal herb’s immunological ability. Notable items in this group include *Astragalus membranaceus*, *Angelica sinensis,* and fungal varieties [2,73].

### 4.2. Activity of Macrophages: [74]

engulfing ability andanticancer activities.

Polysaccharides in medicinal plants have been of major interest, and many examples are related to fungal species. Triterpenes, saponins, and nucleotides are other components of great interests, exemplified by *Panax ginseng.*

### 4.3. Immune Cells—T Cells, B Cells, Natural Killer (NK) Cells: [75,76]

T cells’ responses give good pictures of anti-infection and anticancer influences,B cells could be studied on their responses to polysaccharide stimulation, andNK cell’s direct cytotoxic effects have been observed.

A large number of medicinal herbs, exemplified by *Astragalus membranaceus, Hedyotis* species, *Acanthopanax senticosus,* and *Ledebouriella divaricata* have been found influential.

### 4.4. Cytokines, Interleukin Production [77,78]

Herbal extracts mainly act on the macrophages and related cells. Cellular activities, however, produce cytokines and chemokines, which have far-reaching immunological effects. Studies, therefore, include their productions accepted as extended cellular effects. Selected herbs are those utilized in cellular studies [74]. Other studies include analysis of interferons, TNF-α, etc. [78,79,80].

## 5. Discussion

Medicinal plants have been used for centuries for the treatment of various diseases in many parts of the world. The utility of natural products as important resources for the discovery of new and safe drugs for human healthcare apparently is still important and is ongoing. It has been estimated that about 65% of the world’s population relies on plant-derived medicines for treating various diseases [81]. The demand for natural products has been growing during the past few decades. Herbal dietary supplements continued to experience strong sales growth in the United States in 2019, with an increase by 8.6% [82]. While a recent pilot survey in the UK showed that over 80% of participants used medicinal plants for multiple health benefits [83].

Cancer is one of the leading causes of death in the world, and it has caused the death of 9.9 million people in 2020 [84]. Cancers of the lung, breast, colorectum, liver, stomach, and prostate were listed as the top six cancers in causing mortality [84]. The use of complementary and alternative medicines (CAMs) by patients suffering from cancers is quite well documented [85]. Nowadays, many cancer patients use herbal medicines and natural products derived from plants or mushrooms to combat cancer, to strengthen the immune system, and to counter some possible side effects of the conventional treatment [86]. Approximately 60% of the anticancer agents that are currently available for clinical use or are in late stages of clinical trials are derived from natural products [87].

According to previous reviews, the most frequently used CAM modalities in Europe are herbal medicines [88]. In fact, herbal medicine is at the forefront of traditional medicine in Germany and Western Europe [89]. An Italian survey revealed that herbal medicine is frequently employed to improve quality of life [90]. Recent studies also showed that dissatisfaction with conventional medicine is the most important reason for the preferred use of herbal medicine in Germany [91]. In fact, a similar scenario has occurred with cancer patients worldwide. The use of Chinese herbal medicines (CHMs) as an adjuvant to cancer treatment is more frequent among cancer patients in Chinese communities, such as in China, Hong Kong, and Taiwan [59,92,93,94]. Plentiful clinical studies can be found on the efficacies of traditional Chinese medicines (TCMs), either combined with conventional cancer therapies or used alone, in cancer patients. The conclusions drawn from most of these studies suggest that, for cancer patients, TCM can be useful and effective, helping to control the disease, prolonging survival time, alleviating side effects of chemotherapy and radiotherapy, and improving the quality of life [95]. One systematic review summarized that traditional Chinese medicine preparations, combined with chemotherapy, may improve the objective response rates and disease control rates more than chemotherapy alone [96]. Throughout these years, pharmacognosists, pharmacologists, and biochemists are anticipating more and more scientific evidence on the efficacies of medicinal herbs, to be gained through modern pharmacological research approaches. This is also the mission of our institute. In this review, the preclinical studies of several medicinal herbs and mushrooms are chosen as examples. Their applications as adjuvant therapeutic agents in the management of cancer, particularly in the areas related to immunological support, are expected to gain increasing enthusiasm in future.

During the current COVID-19 pandemic, our research interests on the immunomodulatory effects of medicinal herbs have shifted to include their use as immuno-boosters for innate-immunological defense. Our efforts could be discussed in subsequent reports.

The academic editor of this journal, while reviewing this manuscript, brought forward the concern about the quality issues of the herbal materials and the methodological challenges associated with clinical studies. Indeed, the two issues deserve a separate review to reach a thorough recommendation. We tackle the quality issue through a stringent process: consisting of firstly acquiring the selected herbs from the most reliable seller, followed by comprehensive authentications, from gross examinations to chemical scrutiny; then storing the information in our regional herbal data authority to ensure that future repeated research material could be cross matched to prove identical criteria. With regard to clinical studies, we insist on taking reference from the clinical trial procedures so that safety comes first, followed by dosage identifications before the proper clinical trial could be planned.

## Figures and Tables

**Figure 1 molecules-26-02173-f001:**
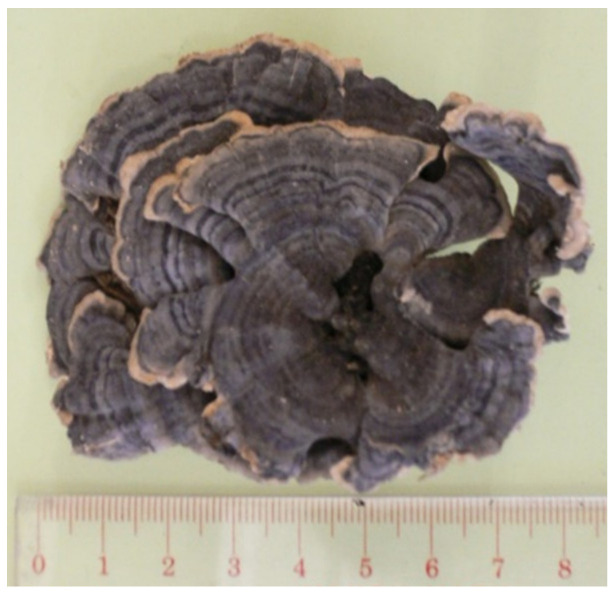
*Coriolus versicolor* (Yunzhi).

**Table 1 molecules-26-02173-t001:** Medicinal plants and mushrooms with anticancer and immunomodulatory effects described in the text.

Name	Photos of Plants/Mushrooms	Main Habitat	Medicinal Use	Main Active Chemical Component	Our Main Research Findings	References
(1) *Agaricus blazei*	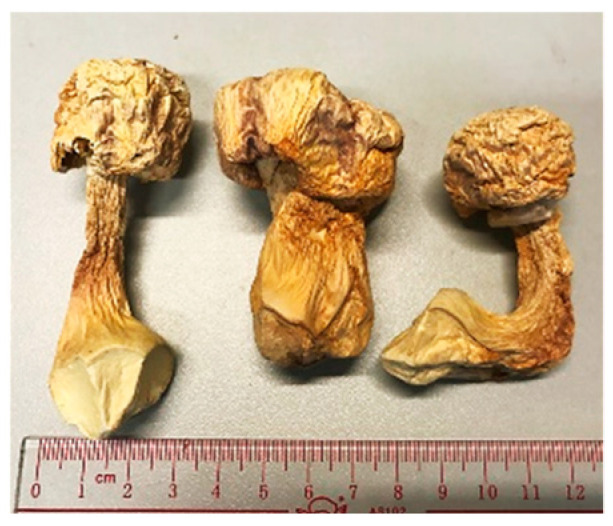	Brazil, East Asia	Cancer, inflammatory/allergy conditions	Crude extracts	Anti-cancer effects	[12,13]
(2) *Ganoderma lucidum*	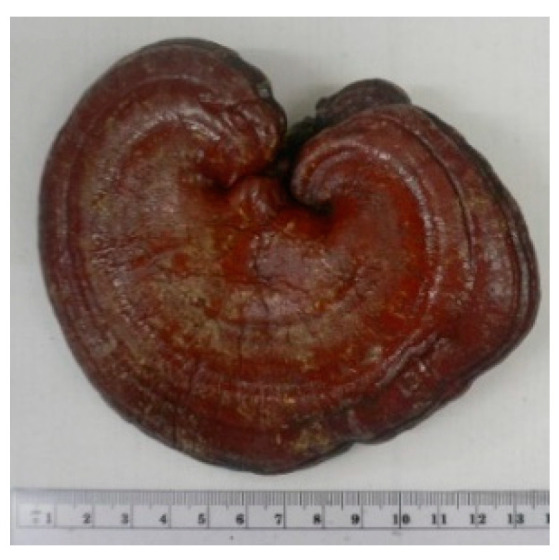	China, East Asia	Longevity, cancer	Triterpenes, polysaccharides	Immunomodulatory effects	[18,19,20,21,22,23,24]
*Ganoderma sinense*	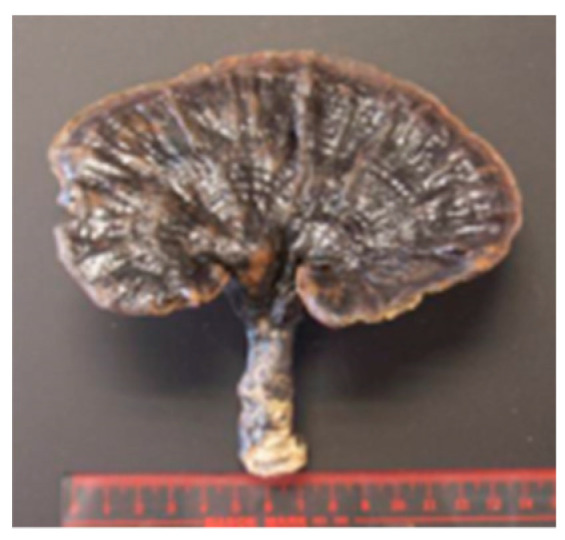					
(3) *Rubinoboletus ballouii*	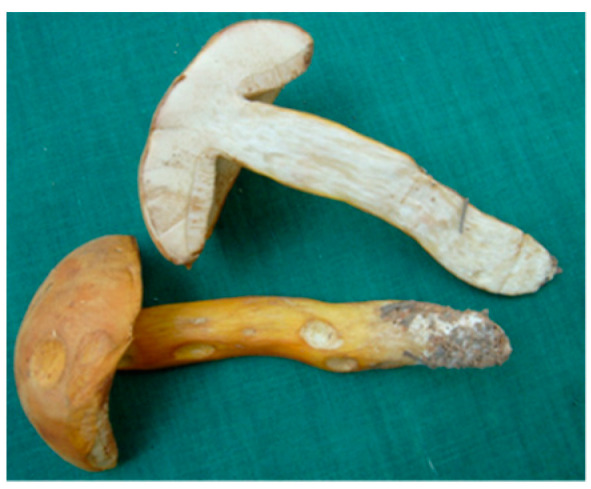	Southwest China	Folk practice	Polysaccharides	Anti-inflammation, immunomodulatory effects	[8,9]
(4) *Acanthopanax senticosus*	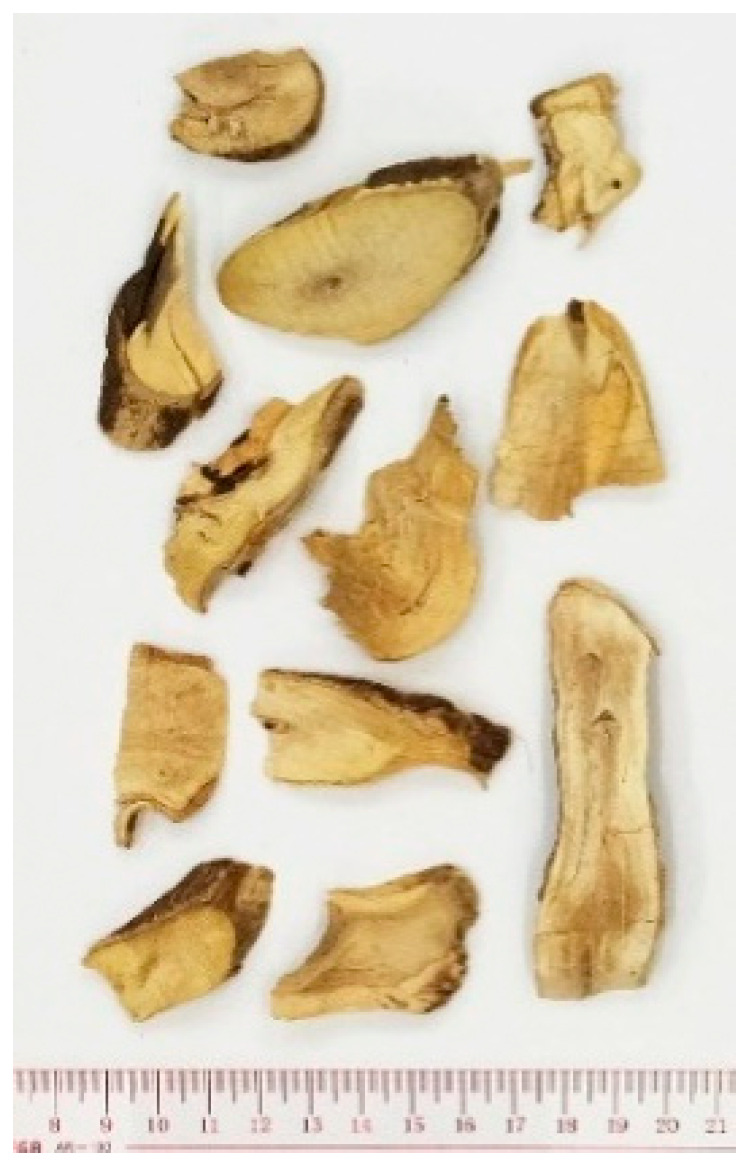	North China	Cancer, rejuvenation	Crude extracts	Immunomodulatory effects	[32]
(5) *Astragalus membranaceus*	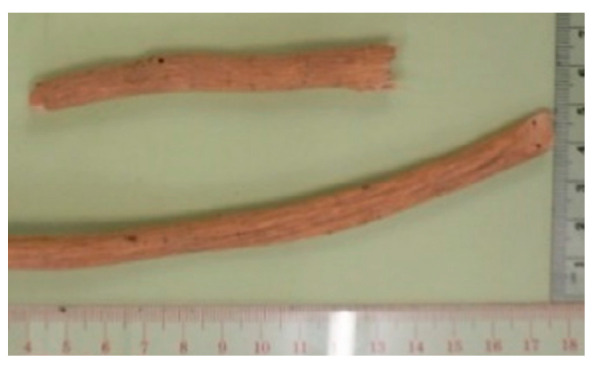	North China	Rejuvenation, infection	Polysaccharides	Immunomodulatory effects	[3,4,5]
(6) *Curcuma longa*	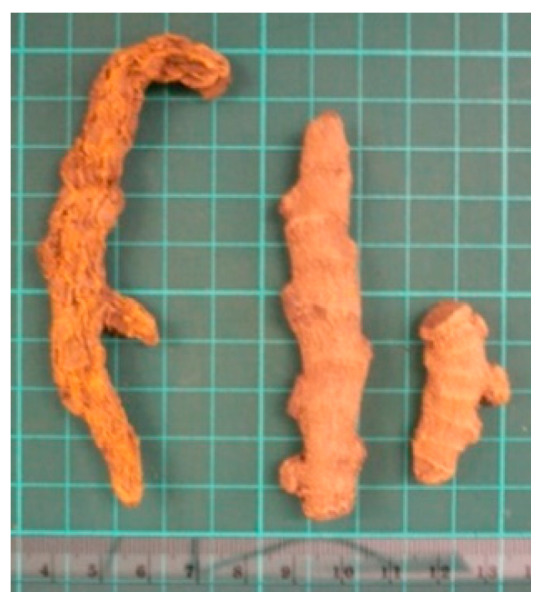	India, South China	Anti-aging, antioxidant, cancer	Crude extracts	Anti-inflammation, anti-angiogenesis, immunomodulatory effects	[6,7,33,34,35,36,37,38]
(7) *Hedyotis diffusa*	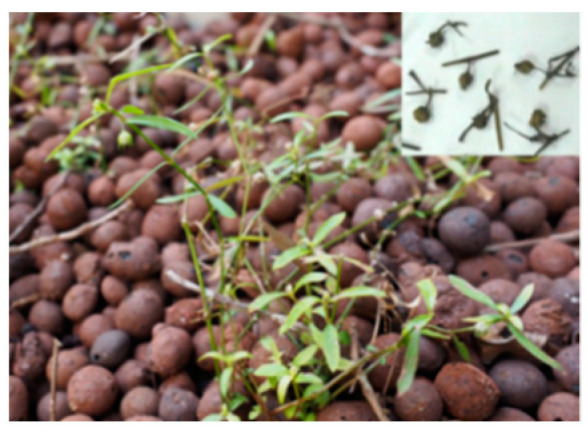	South China	Cancer	Crude extracts	Anti-angiogenesis, anti-cancer	[39,40,41]
(8) *Scutellaria barbata*	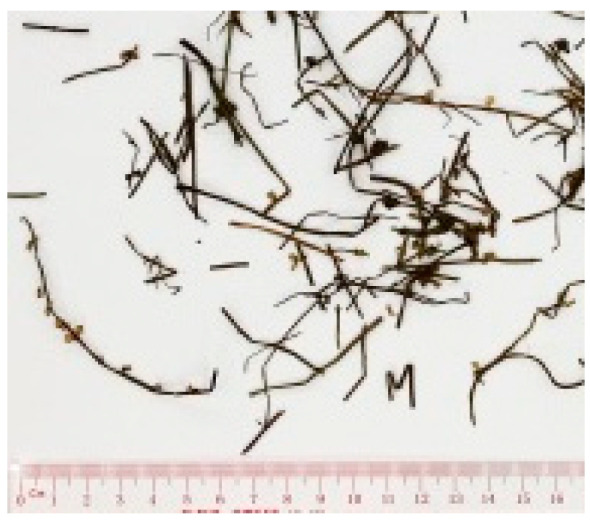	China	Infection, allergy	Crude extracts.Pheophorbide a	Anti-inflammation, detoxication, anti-cancer	[42,43,44,45]

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
