# Peer review of "Medicinal Plants and Mushrooms with Immunomodulatory and Anticancer Properties—A Review on Hong Kong’s Experience"

_molecules, 2021, doi:10.3390/molecules26082173_

Round 1
Reviewer 1 Report
The document under consideration is a rather simplistic compilation of existing literature. However, the information presented is mere cataloguing, rather than critical appraisal of the cited literature. The review does not provide much in the form of specific details.
Section 2 lacks pertinent quantitative details of doses and responses.
Section 3 has some details of dose and statistical significance of the results. However, there is no discussion of the clinical significance of the results. Statistical significance alone is not sufficient in the clinical context. Was the effect clinically worthwhile? Interestingly, these studies are quite old, dating back to 2004 – 2006 and there seems to have been no follow up studies to replicate and expand on these findings in the decade and a half since the initial studies.
English language and style can be improved. For example (the list is NOT exhaustive; highlighted expressions are problematic):
Line 15: “Since Hong Kong people have a special favour given to one medicinal mushroom,
Line 144: “Nevertheless, the immunomodulatory effect of RB worth further investigation.”
Line 193 “…..we testified our hypothesis…” [a hypothesis is tested, not testified]
Line 195-196: “Results from in vitro Caco-2 cell monolayer study [46] and in vivo pharmacokinetic study in mice [47] have proven this hypothesis.” [This expression is unscientific].
Line 336: “Thymes”
Author Response
The document under consideration is a rather simplistic compilation of existing literature. However, the information presented is mere cataloguing, rather than critical appraisal of the cited literature. The review does not provide much in the form of specific details.
Response: Thank you for the comment. This review aimed at giving the results of our exploration on the biological activities of selected medicinal herbs, which were investigated in our Institute.
Section 2 lacks pertinent quantitative details of doses and responses.
Response: Thank you for the comment. The quantitative details of doses and responses have now been added in the revised manuscript.
Section 3 has some details of dose and statistical significance of the results. However, there is no discussion of the clinical significance of the results. Statistical significance alone is not sufficient in the clinical context. Was the effect clinically worthwhile? Interestingly, these studies are quite old, dating back to 2004 – 2006 and there seems to have been no follow up studies to replicate and expand on these findings in the decade and a half since the initial studies.
Response:
Coriolus and salvia capsule were used in patients with nasopharyngeal carcinoma and breast cancer, after their completion of cancer treatment. The studies were meant to explore the changes in some aspects of the immunological data that might favour cancer control. The patients were not followed up but our relevant statistical evaluation was in support of the positive effects (p=<0.05). <This explanation also responds to Reviewer 4’s question 6>
English language and style can be improved. For example (the list is NOT exhaustive; highlighted expressions are problematic):
Line 15: “Since Hong Kong people have a special favour given to one medicinal mushroom,
Response:
Revisions have been made in the text.
Line 144: “Nevertheless, the immunomodulatory effect of RB worth further investigation.”
Response: Thank you for the comment. The sentence has now been revised.
Line 193 “…..we testified our hypothesis…” [a hypothesis is tested, not testified]
Response: The word has now been corrected.
Line 195-196: “Results from in vitro Caco-2 cell monolayer study [46] and in vivo pharmacokinetic study in mice [47] have proven this hypothesis.” [This expression is unscientific].
Response: The sentence has now been revised.
Line 336: “Thymes”
Response: The word has now been corrected.
The document under consideration is a rather simplistic compilation of existing literature. However, the information presented is mere cataloguing, rather than critical appraisal of the cited literature. The review does not provide much in the form of specific details.
Response: Thank you for the comment. This review aimed at giving the results of our exploration on the biological activities of selected medicinal herbs, which were investigated in our Institute.
Section 2 lacks pertinent quantitative details of doses and responses.
Response: Thank you for the comment. The quantitative details of doses and responses have now been added in the revised manuscript.
Section 3 has some details of dose and statistical significance of the results. However, there is no discussion of the clinical significance of the results. Statistical significance alone is not sufficient in the clinical context. Was the effect clinically worthwhile? Interestingly, these studies are quite old, dating back to 2004 – 2006 and there seems to have been no follow up studies to replicate and expand on these findings in the decade and a half since the initial studies.
Response:
Coriolus and salvia capsule were used in patients with nasopharyngeal carcinoma and breast cancer, after their completion of cancer treatment. The studies were meant to explore the changes in some aspects of the immunological data that might favour cancer control. The patients were not followed up but our relevant statistical evaluation was in support of the positive effects (p=<0.05). <This explanation also responds to Reviewer 4’s question 6>
English language and style can be improved. For example (the list is NOT exhaustive; highlighted expressions are problematic):
Line 15: “Since Hong Kong people have a special favour given to one medicinal mushroom,
Response:
Revisions have been made in the text.
Line 144: “Nevertheless, the immunomodulatory effect of RB worth further investigation.”
Response: Thank you for the comment. The sentence has now been revised.
Line 193 “…..we testified our hypothesis…” [a hypothesis is tested, not testified]
Response: The word has now been corrected.
Line 195-196: “Results from in vitro Caco-2 cell monolayer study [46] and in vivo pharmacokinetic study in mice [47] have proven this hypothesis.” [This expression is unscientific].
Response: The sentence has now been revised.
Line 336: “Thymes”
Response: The word has now been corrected.
The document under consideration is a rather simplistic compilation of existing literature. However, the information presented is mere cataloguing, rather than critical appraisal of the cited literature. The review does not provide much in the form of specific details.
Response: Thank you for the comment. This review aimed at giving the results of our exploration on the biological activities of selected medicinal herbs, which were investigated in our Institute.
Section 2 lacks pertinent quantitative details of doses and responses.
Response: Thank you for the comment. The quantitative details of doses and responses have now been added in the revised manuscript.
Section 3 has some details of dose and statistical significance of the results. However, there is no discussion of the clinical significance of the results. Statistical significance alone is not sufficient in the clinical context. Was the effect clinically worthwhile? Interestingly, these studies are quite old, dating back to 2004 – 2006 and there seems to have been no follow up studies to replicate and expand on these findings in the decade and a half since the initial studies.
Response:
Coriolus and salvia capsule were used in patients with nasopharyngeal carcinoma and breast cancer, after their completion of cancer treatment. The studies were meant to explore the changes in some aspects of the immunological data that might favour cancer control. The patients were not followed up but our relevant statistical evaluation was in support of the positive effects (p=<0.05). <This explanation also responds to Reviewer 4’s question 6>
English language and style can be improved. For example (the list is NOT exhaustive; highlighted expressions are problematic):
Line 15: “Since Hong Kong people have a special favour given to one medicinal mushroom,
Response:
Revisions have been made in the text.
Line 144: “Nevertheless, the immunomodulatory effect of RB worth further investigation.”
Response: Thank you for the comment. The sentence has now been revised.
Line 193 “…..we testified our hypothesis…” [a hypothesis is tested, not testified]
Response: The word has now been corrected.
Line 195-196: “Results from in vitro Caco-2 cell monolayer study [46] and in vivo pharmacokinetic study in mice [47] have proven this hypothesis.” [This expression is unscientific].
Response: The sentence has now been revised.
Line 336: “Thymes”
Response: The word has now been corrected.
The document under consideration is a rather simplistic compilation of existing literature. However, the information presented is mere cataloguing, rather than critical appraisal of the cited literature. The review does not provide much in the form of specific details.
Response: Thank you for the comment. This review aimed at giving the results of our exploration on the biological activities of selected medicinal herbs, which were investigated in our Institute.
Section 2 lacks pertinent quantitative details of doses and responses.
Response: Thank you for the comment. The quantitative details of doses and responses have now been added in the revised manuscript.
Section 3 has some details of dose and statistical significance of the results. However, there is no discussion of the clinical significance of the results. Statistical significance alone is not sufficient in the clinical context. Was the effect clinically worthwhile? Interestingly, these studies are quite old, dating back to 2004 – 2006 and there seems to have been no follow up studies to replicate and expand on these findings in the decade and a half since the initial studies.
Response:
Coriolus and salvia capsule were used in patients with nasopharyngeal carcinoma and breast cancer, after their completion of cancer treatment. The studies were meant to explore the changes in some aspects of the immunological data that might favour cancer control. The patients were not followed up but our relevant statistical evaluation was in support of the positive effects (p=<0.05). <This explanation also responds to Reviewer 4’s question 6>
English language and style can be improved. For example (the list is NOT exhaustive; highlighted expressions are problematic):
Line 15: “Since Hong Kong people have a special favour given to one medicinal mushroom,
Response:
Revisions have been made in the text.
Line 144: “Nevertheless, the immunomodulatory effect of RB worth further investigation.”
Response: Thank you for the comment. The sentence has now been revised.
Line 193 “…..we testified our hypothesis…” [a hypothesis is tested, not testified]
Response: The word has now been corrected.
Line 195-196: “Results from in vitro Caco-2 cell monolayer study [46] and in vivo pharmacokinetic study in mice [47] have proven this hypothesis.” [This expression is unscientific].
Response: The sentence has now been revised.
Line 336: “Thymes”
Response: The word has now been corrected.
The document under consideration is a rather simplistic compilation of existing literature. However, the information presented is mere cataloguing, rather than critical appraisal of the cited literature. The review does not provide much in the form of specific details.
Response: Thank you for the comment. This review aimed at giving the results of our exploration on the biological activities of selected medicinal herbs, which were investigated in our Institute.
Section 2 lacks pertinent quantitative details of doses and responses.
Response: Thank you for the comment. The quantitative details of doses and responses have now been added in the revised manuscript.
Section 3 has some details of dose and statistical significance of the results. However, there is no discussion of the clinical significance of the results. Statistical significance alone is not sufficient in the clinical context. Was the effect clinically worthwhile? Interestingly, these studies are quite old, dating back to 2004 – 2006 and there seems to have been no follow up studies to replicate and expand on these findings in the decade and a half since the initial studies.
Response:
Coriolus and salvia capsule were used in patients with nasopharyngeal carcinoma and breast cancer, after their completion of cancer treatment. The studies were meant to explore the changes in some aspects of the immunological data that might favour cancer control. The patients were not followed up but our relevant statistical evaluation was in support of the positive effects (p=<0.05). <This explanation also responds to Reviewer 4’s question 6>
English language and style can be improved. For example (the list is NOT exhaustive; highlighted expressions are problematic):
Line 15: “Since Hong Kong people have a special favour given to one medicinal mushroom,
Response:
Revisions have been made in the text.
Line 144: “Nevertheless, the immunomodulatory effect of RB worth further investigation.”
Response: Thank you for the comment. The sentence has now been revised.
Line 193 “…..we testified our hypothesis…” [a hypothesis is tested, not testified]
Response: The word has now been corrected.
Line 195-196: “Results from in vitro Caco-2 cell monolayer study [46] and in vivo pharmacokinetic study in mice [47] have proven this hypothesis.” [This expression is unscientific].
Response: The sentence has now been revised.
Line 336: “Thymes”
Response: The word has now been corrected.
The document under consideration is a rather simplistic compilation of existing literature. However, the information presented is mere cataloguing, rather than critical appraisal of the cited literature. The review does not provide much in the form of specific details.
Response: Thank you for the comment. This review aimed at giving the results of our exploration on the biological activities of selected medicinal herbs, which were investigated in our Institute.
Section 2 lacks pertinent quantitative details of doses and responses.
Response: Thank you for the comment. The quantitative details of doses and responses have now been added in the revised manuscript.
Section 3 has some details of dose and statistical significance of the results. However, there is no discussion of the clinical significance of the results. Statistical significance alone is not sufficient in the clinical context. Was the effect clinically worthwhile? Interestingly, these studies are quite old, dating back to 2004 – 2006 and there seems to have been no follow up studies to replicate and expand on these findings in the decade and a half since the initial studies.
Response:
Coriolus and salvia capsule were used in patients with nasopharyngeal carcinoma and breast cancer, after their completion of cancer treatment. The studies were meant to explore the changes in some aspects of the immunological data that might favour cancer control. The patients were not followed up but our relevant statistical evaluation was in support of the positive effects (p=<0.05). <This explanation also responds to Reviewer 4’s question 6>
English language and style can be improved. For example (the list is NOT exhaustive; highlighted expressions are problematic):
Line 15: “Since Hong Kong people have a special favour given to one medicinal mushroom,
Response:
Revisions have been made in the text.
Line 144: “Nevertheless, the immunomodulatory effect of RB worth further investigation.”
Response: Thank you for the comment. The sentence has now been revised.
Line 193 “…..we testified our hypothesis…” [a hypothesis is tested, not testified]
Response: The word has now been corrected.
Line 195-196: “Results from in vitro Caco-2 cell monolayer study [46] and in vivo pharmacokinetic study in mice [47] have proven this hypothesis.” [This expression is unscientific].
Response: The sentence has now been revised.
Line 336: “Thymes”
Response: The word has now been corrected.
The document under consideration is a rather simplistic compilation of existing literature. However, the information presented is mere cataloguing, rather than critical appraisal of the cited literature. The review does not provide much in the form of specific details.
Response: Thank you for the comment. This review aimed at giving the results of our exploration on the biological activities of selected medicinal herbs, which were investigated in our Institute.
Section 2 lacks pertinent quantitative details of doses and responses.
Response: Thank you for the comment. The quantitative details of doses and responses have now been added in the revised manuscript.
Section 3 has some details of dose and statistical significance of the results. However, there is no discussion of the clinical significance of the results. Statistical significance alone is not sufficient in the clinical context. Was the effect clinically worthwhile? Interestingly, these studies are quite old, dating back to 2004 – 2006 and there seems to have been no follow up studies to replicate and expand on these findings in the decade and a half since the initial studies.
Response:
Coriolus and salvia capsule were used in patients with nasopharyngeal carcinoma and breast cancer, after their completion of cancer treatment. The studies were meant to explore the changes in some aspects of the immunological data that might favour cancer control. The patients were not followed up but our relevant statistical evaluation was in support of the positive effects (p=<0.05). <This explanation also responds to Reviewer 4’s question 6>
English language and style can be improved. For example (the list is NOT exhaustive; highlighted expressions are problematic):
Line 15: “Since Hong Kong people have a special favour given to one medicinal mushroom,
Response:
Revisions have been made in the text.
Line 144: “Nevertheless, the immunomodulatory effect of RB worth further investigation.”
Response: Thank you for the comment. The sentence has now been revised.
Line 193 “…..we testified our hypothesis…” [a hypothesis is tested, not testified]
Response: The word has now been corrected.
Line 195-196: “Results from in vitro Caco-2 cell monolayer study [46] and in vivo pharmacokinetic study in mice [47] have proven this hypothesis.” [This expression is unscientific].
Response: The sentence has now been revised.
Line 336: “Thymes”
Response: The word has now been corrected.
The document under consideration is a rather simplistic compilation of existing literature. However, the information presented is mere cataloguing, rather than critical appraisal of the cited literature. The review does not provide much in the form of specific details.
Response: Thank you for the comment. This review aimed at giving the results of our exploration on the biological activities of selected medicinal herbs, which were investigated in our Institute.
Section 2 lacks pertinent quantitative details of doses and responses.
Response: Thank you for the comment. The quantitative details of doses and responses have now been added in the revised manuscript.
Section 3 has some details of dose and statistical significance of the results. However, there is no discussion of the clinical significance of the results. Statistical significance alone is not sufficient in the clinical context. Was the effect clinically worthwhile? Interestingly, these studies are quite old, dating back to 2004 – 2006 and there seems to have been no follow up studies to replicate and expand on these findings in the decade and a half since the initial studies.
Response:
Coriolus and salvia capsule were used in patients with nasopharyngeal carcinoma and breast cancer, after their completion of cancer treatment. The studies were meant to explore the changes in some aspects of the immunological data that might favour cancer control. The patients were not followed up but our relevant statistical evaluation was in support of the positive effects (p=<0.05). <This explanation also responds to Reviewer 4’s question 6>
English language and style can be improved. For example (the list is NOT exhaustive; highlighted expressions are problematic):
Line 15: “Since Hong Kong people have a special favour given to one medicinal mushroom,
Response:
Revisions have been made in the text.
Line 144: “Nevertheless, the immunomodulatory effect of RB worth further investigation.”
Response: Thank you for the comment. The sentence has now been revised.
Line 193 “…..we testified our hypothesis…” [a hypothesis is tested, not testified]
Response: The word has now been corrected.
Line 195-196: “Results from in vitro Caco-2 cell monolayer study [46] and in vivo pharmacokinetic study in mice [47] have proven this hypothesis.” [This expression is unscientific].
Response: The sentence has now been revised.
Line 336: “Thymes”
Response: The word has now been corrected.
The document under consideration is a rather simplistic compilation of existing literature. However, the information presented is mere cataloguing, rather than critical appraisal of the cited literature. The review does not provide much in the form of specific details.
Response: Thank you for the comment. This review aimed at giving the results of our exploration on the biological activities of selected medicinal herbs, which were investigated in our Institute.
Section 2 lacks pertinent quantitative details of doses and responses.
Response: Thank you for the comment. The quantitative details of doses and responses have now been added in the revised manuscript.
Section 3 has some details of dose and statistical significance of the results. However, there is no discussion of the clinical significance of the results. Statistical significance alone is not sufficient in the clinical context. Was the effect clinically worthwhile? Interestingly, these studies are quite old, dating back to 2004 – 2006 and there seems to have been no follow up studies to replicate and expand on these findings in the decade and a half since the initial studies.
Response:
Coriolus and salvia capsule were used in patients with nasopharyngeal carcinoma and breast cancer, after their completion of cancer treatment. The studies were meant to explore the changes in some aspects of the immunological data that might favour cancer control. The patients were not followed up but our relevant statistical evaluation was in support of the positive effects (p=<0.05). <This explanation also responds to Reviewer 4’s question 6>
English language and style can be improved. For example (the list is NOT exhaustive; highlighted expressions are problematic):
Line 15: “Since Hong Kong people have a special favour given to one medicinal mushroom,
Response:
Revisions have been made in the text.
Line 144: “Nevertheless, the immunomodulatory effect of RB worth further investigation.”
Response: Thank you for the comment. The sentence has now been revised.
Line 193 “…..we testified our hypothesis…” [a hypothesis is tested, not testified]
Response: The word has now been corrected.
Line 195-196: “Results from in vitro Caco-2 cell monolayer study [46] and in vivo pharmacokinetic study in mice [47] have proven this hypothesis.” [This expression is unscientific].
Response: The sentence has now been revised.
Line 336: “Thymes”
Response: The word has now been corrected.
The document under consideration is a rather simplistic compilation of existing literature. However, the information presented is mere cataloguing, rather than critical appraisal of the cited literature. The review does not provide much in the form of specific details.
Response: Thank you for the comment. This review aimed at giving the results of our exploration on the biological activities of selected medicinal herbs, which were investigated in our Institute.
Section 2 lacks pertinent quantitative details of doses and responses.
Response: Thank you for the comment. The quantitative details of doses and responses have now been added in the revised manuscript.
Section 3 has some details of dose and statistical significance of the results. However, there is no discussion of the clinical significance of the results. Statistical significance alone is not sufficient in the clinical context. Was the effect clinically worthwhile? Interestingly, these studies are quite old, dating back to 2004 – 2006 and there seems to have been no follow up studies to replicate and expand on these findings in the decade and a half since the initial studies.
Response:
Coriolus and salvia capsule were used in patients with nasopharyngeal carcinoma and breast cancer, after their completion of cancer treatment. The studies were meant to explore the changes in some aspects of the immunological data that might favour cancer control. The patients were not followed up but our relevant statistical evaluation was in support of the positive effects (p=<0.05). <This explanation also responds to Reviewer 4’s question 6>
English language and style can be improved. For example (the list is NOT exhaustive; highlighted expressions are problematic):
Line 15: “Since Hong Kong people have a special favour given to one medicinal mushroom,
Response:
Revisions have been made in the text.
Line 144: “Nevertheless, the immunomodulatory effect of RB worth further investigation.”
Response: Thank you for the comment. The sentence has now been revised.
Line 193 “…..we testified our hypothesis…” [a hypothesis is tested, not testified]
Response: The word has now been corrected.
Line 195-196: “Results from in vitro Caco-2 cell monolayer study [46] and in vivo pharmacokinetic study in mice [47] have proven this hypothesis.” [This expression is unscientific].
Response: The sentence has now been revised.
Line 336: “Thymes”
Response: The word has now been corrected.
The document under consideration is a rather simplistic compilation of existing literature. However, the information presented is mere cataloguing, rather than critical appraisal of the cited literature. The review does not provide much in the form of specific details.
Response: Thank you for the comment. This review aimed at giving the results of our exploration on the biological activities of selected medicinal herbs, which were investigated in our Institute.
Section 2 lacks pertinent quantitative details of doses and responses.
Response: Thank you for the comment. The quantitative details of doses and responses have now been added in the revised manuscript.
Section 3 has some details of dose and statistical significance of the results. However, there is no discussion of the clinical significance of the results. Statistical significance alone is not sufficient in the clinical context. Was the effect clinically worthwhile? Interestingly, these studies are quite old, dating back to 2004 – 2006 and there seems to have been no follow up studies to replicate and expand on these findings in the decade and a half since the initial studies.
Response:
Coriolus and salvia capsule were used in patients with nasopharyngeal carcinoma and breast cancer, after their completion of cancer treatment. The studies were meant to explore the changes in some aspects of the immunological data that might favour cancer control. The patients were not followed up but our relevant statistical evaluation was in support of the positive effects (p=<0.05). <This explanation also responds to Reviewer 4’s question 6>
English language and style can be improved. For example (the list is NOT exhaustive; highlighted expressions are problematic):
Line 15: “Since Hong Kong people have a special favour given to one medicinal mushroom,
Response:
Revisions have been made in the text.
Line 144: “Nevertheless, the immunomodulatory effect of RB worth further investigation.”
Response: Thank you for the comment. The sentence has now been revised.
Line 193 “…..we testified our hypothesis…” [a hypothesis is tested, not testified]
Response: The word has now been corrected.
Line 195-196: “Results from in vitro Caco-2 cell monolayer study [46] and in vivo pharmacokinetic study in mice [47] have proven this hypothesis.” [This expression is unscientific].
Response: The sentence has now been revised.
Line 336: “Thymes”
Response: The word has now been corrected.
Reviewer 2 Report
Well written manuscript which covers selected medicinal plants and mushrooms. My main concern is the lack of images of the plants/mushrooms and the lack of chemical structures of the main bioactive components from each of the selected plants and mushrooms. More detail would also be useful.
Author Response
Thank you for the comment. Table 1 has been revised and the images of the plants and mushrooms are now added. The chemical structures of the main bioactive components from each of the selected plants and mushrooms can be found in our references provided.
Reviewer 3 Report
Abstract
- Editorial errors should be corrected.
- Introduction
- Sentences too complex (sometimes so long that you don't know what's going on). Linguistic errors: stylistic and grammatical. Therefore, the text is complex and incomprehensible. Understanding the text and the entire message is very difficult. The text requires linguistic correction.
- Plant items - what is it? There is no such term in scientific (and general) literature. Maybe it's the result of an incorrect translation?
- The Hong Experience: Immunological Properties of Medicinal Herbs Used Against Cancer
- Please check linguistic correctness - minor corrections required.
- Row 48 – “They may either directly inhibit the growth of tumor or indirectly exert anti-tumor effect by enhancing the body immune function”. - why is there no literature citation?
- Row 53 - it would be advisable to give Latin species names
- Row 57 - TCM theory - no explanation of what the abbreviation means (it appears here for the first time in the manuscript)
- Table 1 - there is no literature for the described research in the table. The graphic side should also be better.
- The Hong Kong Experience on Clinical Applications
- Row 264 - it would be advisable to explain the PSP abbreviation (it appears here for the first time in the manuscript)
- Figure 1 – “Coriolus Versicolor” should be written: Coriolus versicolor
Figure is too small, poor resolution. The source of the photo is not given
- Row 332-333, 335 - why is every word capitalized?
- Row 334, 342, 348 - shouldn't there be any spaces from the preceding text to make the whole text more readable?
- Row 346 - fungal items? There is no such term in the scientific (and general) literature. Maybe it's the result of an incorrect translation?
- Row 346 - is: “Triterpenes saponins,” should be “Triterpenes, saponins,”
- Row 349-351 - shouldn't a.-c. be bulleted? as above in the text?
- Discussion
- Row 371 – “Cancer is one of the leading causes of death in the world and it has caused the death 370 of 9.9 million people in 2020”. - no literature reference
Author Response
Abstract
Editorial errors should be corrected.
Response:
Thank you for the comments, response:
- Linguistic improvements have been taken care of.
- Citation was provided.
- The Latin species names were provided in subsequent paragraphs.
- “TCM” was converted to “Traditional Chinese Medicine”.
- Table 1 is a simple reminder of the items described under Section 2. Details of relevant research are given under separate items.
- Other typos are corrected accordingly.
- Introduction
Sentences too complex (sometimes so long that you don't know what's going on). Linguistic errors: stylistic and grammatical. Therefore, the text is complex and incomprehensible. Understanding the text and the entire message is very difficult. The text requires linguistic correction.
Plant items - what is it? There is no such term in scientific (and general) literature. Maybe it's the result of an incorrect translation?
Response:
“Plant items” is changed to “Herbal formulation”.
- The Hong Experience: Immunological Properties of Medicinal Herbs Used Against Cancer
Please check linguistic correctness - minor corrections required.
Row 48 – “They may either directly inhibit the growth of tumor or indirectly exert anti-tumor effect by enhancing the body immune function”. - why is there no literature citation?
Response: The sentence is a brief summary made by authors. So no reference was cited.
Row 53 - it would be advisable to give Latin species names
Response: The Latin species names have now been added.
Row 57 - TCM theory - no explanation of what the abbreviation means (it appears here for the first time in the manuscript)
Response: TCM stands for traditional Chinese medicine. This is now clarified in the revised manuscript.
Table 1 - there is no literature for the described research in the table. The graphic side should also be better.
Response: The references have now been added.
- The Hong Kong Experience on Clinical Applications
Row 264 - it would be advisable to explain the PSP abbreviation (it appears here for the first time in the manuscript)
Figure 1 – “Coriolus Versicolor” should be written: Coriolus versicolor
Figure is too small, poor resolution. The source of the photo is not given
Row 332-333, 335 - why is every word capitalized?
Row 334, 342, 348 - shouldn't there be any spaces from the preceding text to make the whole text more readable?
Row 346 - fungal items? There is no such term in the scientific (and general) literature. Maybe it's the result of an incorrect translation?
Row 346 - is: “Triterpenes saponins,” should be “Triterpenes, saponins,”
Row 349-351 - shouldn't a.-c. be bulleted? as above in the text?
Response:
Thank you for the comments and suggestions.
Corrections and amendments have been made according to the comments and suggestion in the relevant rows. (see text)
- Discussion
Row 371 – “Cancer is one of the leading causes of death in the world and it has caused the death 370 of 9.9 million people in 2020”. - no literature reference
Response: The reference has now been added.
Reviewer 4 Report
Opinion related to the paper entitled: Medicinal plants and mushrooms with immunomodulatory and anti-cancer properties – a review of Hong Kong experience.
Cancer diseases are among the more common causes of death, after cardiovascular diseases. Chemotherapeutic or/and radiotherapeutic treatment is associated with a significant weakening of the organism, including immune, defense functions. For this reason it seems attractive, the possibility of using herbs for supportive treatment of cancer, leading to prolongation of life, reduction of side effects, and in some cases, after strengthening the organism, also cure.
A work worth publishing after error correction.
List of comments.
- line 53 – give Latin names instead of turmeric and maitake.
- line 74 – Authors classify Agaricus blazei as an edible mushroom. There are some information related to the toxic effect of this mushroom on liver condition.
- lines 141/ 142 Boletus write in italics.
- line 168/173 Radix Astragali write in italics.
- lines 286 – better give Latin names instead of Yun Zhi and Danshen.
- lines 303-316 – section 3.2 is unclear. It should be explained how the patients were treated and what benefits resulted from the use natural preparations.
- lines 332/333 – why capital letter were used.
- line 336 – should be Thymus.
Author Response
Opinion related to the paper entitled: Medicinal plants and mushrooms with immunomodulatory and anti-cancer properties – a review of Hong Kong experience.
Cancer diseases are among the more common causes of death, after cardiovascular diseases. Chemotherapeutic or/and radiotherapeutic treatment is associated with a significant weakening of the organism, I ncluding immune, defense functions. For this reason it seems attractive, the possibility of using herbs for supportive treatment of cancer, leading to prolongation of life, reduction of side effects, and in some cases, after strengthening the organism, also cure.
A work worth publishing after error correction.
List of comments.
- line 53 – give Latin names instead of turmeric and maitake.
Response: The Latin species names have now been added.
- line 74 – Authors classify Agaricus blazei as an edible mushroom. There are some information related to the toxic effect of this mushroom on liver condition.
Response: Thank you for the comment. We agree that some toxic effects of Agaricus blazei have been reported. However, this mushroom has long been regarded as edible mushroom with medicinal properties since 2002 (Hashimoto et al., 2002; Kaneno et al., 2004; Wang et al., 2013). Thus we classified this mushroom as edible mushroom.
References:
Hashimoto T, Nonaka Y, Minato K, Kawakami S, Mizuno M, Fukuda I, Kanazawa K, Ashida H. Suppressive effect of polysaccharides from the edible and medicinal mushrooms, Lentinus edodes and Agaricus blazei, on the expression of cytochrome P450s in mice. Biosci Biotechnol Biochem. 2002 Jul;66(7):1610-4. doi: 10.1271/bbb.66.1610.
Kaneno R, Fontanari LM, Santos SA, Di Stasi LC, Rodrigues Filho E, Eira AF. Effects of extracts from Brazilian sun-mushroom (Agaricus blazei) on the NK activity and lymphoproliferative responsiveness of Ehrlich tumor-bearing mice. Food Chem Toxicol. 2004 Jun;42(6):909-16. doi: 10.1016/j.fct.2004.01.014.
Wang H, Fu Z, Han C. The medicinal values of culinary-medicinal royal sun mushroom (Agaricus blazei Murrill). Evid Based Complement Alternat Med. 2013;2013:842619. doi: 10.1155/2013/842619.
- lines 141/ 142 Boletus write in italics.
Response: The word has now been corrected.
- line 168/173 Radix Astragali write in italics.
Response: The Chinese herbal names should not be in italics, while the plant name Astragalus membranaceus should be written in italics.
- lines 286 – better give Latin names instead of Yun Zhi and Danshen.
Response: Latin names have been provided.
- lines 303-316 – section 3.2 is unclear. It should be explained how the patients were treated and what benefits resulted from the use natural preparations.
Response:
Thank you for the suggestion. Cancer patients had been given the coriolus and salvia capsules after they have completed standard treatment. They were expected to enjoy better quality of life and more specifically, we critically looked at the immunological effects which would offer quantitative data on clinical benefits. The follow-up periods were short which would not allow longer term studies on clinical benefits.
- lines 332/333 – why capital letter were used.
Response:
The typing error has been corrected.
- line 336 – should be Thymus.
Response: The word has now been corrected.
Opinion related to the paper entitled: Medicinal plants and mushrooms with immunomodulatory and anti-cancer properties – a review of Hong Kong experience.
Cancer diseases are among the more common causes of death, after cardiovascular diseases. Chemotherapeutic or/and radiotherapeutic treatment is associated with a significant weakening of the organism, I ncluding immune, defense functions. For this reason it seems attractive, the possibility of using herbs for supportive treatment of cancer, leading to prolongation of life, reduction of side effects, and in some cases, after strengthening the organism, also cure.
A work worth publishing after error correction.
List of comments.
- line 53 – give Latin names instead of turmeric and maitake.
Response: The Latin species names have now been added.
- line 74 – Authors classify Agaricus blazei as an edible mushroom. There are some information related to the toxic effect of this mushroom on liver condition.
Response: Thank you for the comment. We agree that some toxic effects of Agaricus blazei have been reported. However, this mushroom has long been regarded as edible mushroom with medicinal properties since 2002 (Hashimoto et al., 2002; Kaneno et al., 2004; Wang et al., 2013). Thus we classified this mushroom as edible mushroom.
References:
Hashimoto T, Nonaka Y, Minato K, Kawakami S, Mizuno M, Fukuda I, Kanazawa K, Ashida H. Suppressive effect of polysaccharides from the edible and medicinal mushrooms, Lentinus edodes and Agaricus blazei, on the expression of cytochrome P450s in mice. Biosci Biotechnol Biochem. 2002 Jul;66(7):1610-4. doi: 10.1271/bbb.66.1610.
Kaneno R, Fontanari LM, Santos SA, Di Stasi LC, Rodrigues Filho E, Eira AF. Effects of extracts from Brazilian sun-mushroom (Agaricus blazei) on the NK activity and lymphoproliferative responsiveness of Ehrlich tumor-bearing mice. Food Chem Toxicol. 2004 Jun;42(6):909-16. doi: 10.1016/j.fct.2004.01.014.
Wang H, Fu Z, Han C. The medicinal values of culinary-medicinal royal sun mushroom (Agaricus blazei Murrill). Evid Based Complement Alternat Med. 2013;2013:842619. doi: 10.1155/2013/842619.
- lines 141/ 142 Boletus write in italics.
Response: The word has now been corrected.
- line 168/173 Radix Astragali write in italics.
Response: The Chinese herbal names should not be in italics, while the plant name Astragalus membranaceus should be written in italics.
- lines 286 – better give Latin names instead of Yun Zhi and Danshen.
Response: Latin names have been provided.
- lines 303-316 – section 3.2 is unclear. It should be explained how the patients were treated and what benefits resulted from the use natural preparations.
Response:
Thank you for the suggestion. Cancer patients had been given the coriolus and salvia capsules after they have completed standard treatment. They were expected to enjoy better quality of life and more specifically, we critically looked at the immunological effects which would offer quantitative data on clinical benefits. The follow-up periods were short which would not allow longer term studies on clinical benefits.
- lines 332/333 – why capital letter were used.
Response:
The typing error has been corrected.
- line 336 – should be Thymus.
Response: The word has now been corrected.
Round 2
Reviewer 1 Report
The authors have addressed the several minor points raised, in terms of typos, language and providing some details that were missing in the earlier version. However, the three important issues (noted below) that have not been resolved compel me to recommend ‘Rejection’ of this manuscript.
- the information presented is mere cataloguing, rather than critical appraisal of the cited literature.
- Section 3: these studies are quite old, dating back to 2004 – 2006 and there seems to have been no follow up studies to replicate and expand on these findings in the decade and a half since the initial studies. These results are more than a decade old and are of little relevance in the absence of any updates.
- Statistical significance alone is not sufficient in the clinical context. What was the ‘effect size’? Was the effect clinically worthwhile? Authors’ reiteration that the immunological parameters showed statistically significant results does not throw any light on its clinical importance (significance).